# Does a Polycentric Spatial Structure Help to Reduce Industry Emissions?

**DOI:** 10.3390/ijerph19138167

**Published:** 2022-07-03

**Authors:** Shuaishuai Han, Changhong Miao

**Affiliations:** 1Key Research Institute of Yellow River Civilization and Sustainable Development & Collaborative Innovation Center on Yellow River Civilization Jointly Built by Henan Province and Ministry of Education, Henan University, Kaifeng 475001, China; sshan@henu.edu.cn; 2Institute of Eco-Chongming, 20 Cuiniao Rd., Chenjia Zhen, Chongming, Shanghai 202162, China

**Keywords:** polycentricity, firm-level emissions, urban spatial structure, industry, spatial distribution

## Abstract

City planners are increasingly drawn to ways of transforming urban spatial structure as an important strategy for reducing pollutant emissions. As its main contribution, this paper uses firm-level emissions data to quantify impact mechanisms related to factor flow, firm size, and division of labour. We examine the effects of spatial polycentricity on firm-level industrial emissions, using a pooled cross-sectional model, based on emissions data from individual firms in China. We show that, all else being equal, polycentric spatial structures help to reduce the emissions of industrial firms. This finding is not affected by index measures, changes in industrial structure, or city-sample selection. A mechanism analysis shows that polycentric structures not only enhance the emission-reduction effects of factor flow and firm size, but also reduce firm-level emissions by strengthening the urban division of labour. Our findings support the emission-reduction performance of polycentric spatial structures, promoting the integration of city planning and industrial policies that jointly contribute to reducing firm-level emissions and preventing and controlling air pollution.

## 1. Introduction

In the current pattern of urban development in China, the concentration of the population in cities is a fundamental reality. While urbanization improves production efficiency and infrastructure, it also has a highly negative impact on the ecological environment. Air pollution caused by the over-concentration of resources represents a major constraint on China’s sustainable economic development and the wellbeing of its residents. Industrial emissions are the primary source of air pollution [1]; the essence of cities is agglomeration [2]. The question of how to maintain urban expansion while reducing industrial emissions during the agglomeration process has become a hot topic in the field of Chinese urbanization.

Numerous studies on urban agglomeration and air pollution reveal a strong relationship between urban spatial structure and industrial emissions [3]. Agglomeration can improve the environmental technologies used by industry through knowledge spillovers and innovation [4]. However, over-agglomeration also leads to congestion effects and a race to the bottom, increasing industrial emissions. In the trade-off between agglomeration economies and diseconomies, optimizing an agglomeration structure is believed to increase the optimal level of agglomeration [5], reducing industrial emissions while expanding city size.

In city-planning practices, polycentric structures are often used to address urban environmental problems. For example, one measure adopted during the construction of the Tongzhou administrative subcentre in Beijing was to relocate pollution-intensive firms to the suburbs. By influencing the production behaviour of firms, this measure improved environmental quality. The Shanghai Urban Master Plan (2016–2040) recommends using a ‘polycentric and clustered’ spatial structure to achieve a green, low-carbon ecological city. Alongside mega-cities, many medium-sized cities are also developing polycentric models—constructing new city centres and districts to relieve environmental pressure on the main city proper. Although polycentric city-planning practices are developing vigorously, there is no consensus on the extent to which polycentric spatial structures actually reduce industrial emissions. More empirical evidence is needed to confirm this.

The present study links the China Industry Business Performance Database with the China Pollution Database, using 439,493 records of firm-level emissions between 2000 and 2012. Combining these records with data from the China City Statistical Yearbook for the corresponding years has made it possible to carry out a quantitative investigation of the impact of urban spatial polycentricity on industrial firm-level emissions. In contrast with the previous literature on spatial structures and air pollution, the present study aims to make breakthroughs in two areas. First, unlike prior studies, which have examined city- or regional-scale aggregate pollution data, we have used firm-level emissions data to examine the influence of urban spatial structures on the emissions behaviour of individual firms. Such structures can effectively control the impact of firm characteristics on emissions behaviour, helping to eliminate the urban- or regional-scale average effect of firm-level emissions. Second, instead of summarizing impact mechanisms as an agglomeration economy or diseconomy, in line with prior studies, we have conducted a quantitative investigation of the impact of polycentricity on firm-level emissions, mediated by factor flow, firm size, and division of labour.

## 2. Literature Review and Conceptual Hypotheses

### 2.1. The Impact of Polycentric Spatial Structures on Industrial Emissions

Few studies have investigated the link between polycentric spatial structures and industrial emissions. Chen and Zhang (2021) analysed data from Chinese provinces, showing that industrial emissions per unit of output value showed a significant downward trend as the polycentricity of a province’s spatial structure increased [6]. In regard to impact mechanisms, market integration and factor flow are important ways for polycentric spatial structures to reduce industrial emissions.

The nature of a polycentric spatial structure is polycentric agglomeration [7]. The impact of agglomeration on industrial emissions has been well explored. Studies have shown that agglomeration helps to reduce industrial emissions. Otsuka et al. (2014) argued that the agglomeration of related industries in a region could improve production efficiency through specialized production and technology spillover, which in turn would reduce industrial pollutant emissions caused by energy consumption [8]. Lu and Feng (2014) conducted empirical research based on Chinese provincial data; they found that increased population and economic activity agglomeration helped to reduce the intensity of industrial pollutant emissions per unit of output value [9]. Hu et al. (2014) explored the impact of agglomeration on industrial pollutant emissions from the perspective of industrial agglomeration and industrial clusters, finding a negative correlation between industrial agglomeration/clusters and pollutant emissions [10]. Zhang and Dou (2014) suggested that the pollution effect was higher in the central and western regions of China, where the level of agglomeration was relatively low [11].

Other studies have found that agglomeration increases industrial emissions. Ciccone and Hall (1993) found that agglomeration expanded production and increased the level of economic development; as a consequence, areas with higher agglomeration densities often experienced very high industrial pollutant emissions [12]. Verhoef and Nijkamp (2002) argued that industrial agglomeration increased traffic congestion and the cost of product transportation, leading to higher levels of regional industrial pollutant emissions [13]. Chen et al. (2018) observed that, in China, which was undergoing rapid industrial expansion at the time, cities with higher levels of agglomeration tended to have more production inputs and higher levels of industrial emissions [14]. Therefore, the expansion of production scale and the race to the bottom brought about by agglomeration will lead to an increase in industrial emissions.

Inspired by the environmental Kuznets curve, other studies have focused on the nonlinear relationship between agglomeration and industrial emissions. Zhao et al. (2019) discussed the relationship between agglomeration and the emissions of three types of pollutants in China. They found an inverted N-type relationship between agglomeration and emissions of industrial solid waste and sulphur dioxide, and an inverted U-shaped relationship between agglomeration and industrial wastewater discharge [15]. Dinda (2004) showed that, in the early stages of development, increased levels of economic agglomeration are often accompanied by a structural shift towards industrialization, causing gradual increases in the level of industrial pollution; during the later stages of economic development, energy-intensive industries are gradually replaced by clean-energy and knowledge-intensive industries, reducing environmental pollution [16].

Spatial polycentricity is a consequence of decentralization, followed by re-agglomeration [5,17]; this shows a pattern of global decentralization and local agglomeration. Theoretically, polycentricity is associated with high performance because it maintains local economies of agglomeration while eliminating global diseconomies of agglomeration caused by over-agglomeration. Thus, we propose the following hypothesis:

**Hypothesis** **1.**
*Urban spatial polycentricity can significantly reduce the pollutant emissions of industrial firms.*


### 2.2. Impact Mechanisms

The impact of polycentric spatial structures on industrial emissions has not been sufficiently studied. In reviewing the existing literature, we identified the following impact mechanisms: factor flow, firm size, and division of labour.

First, polycentric spatial structures can reduce emissions by increasing factor flows. In a monocentric structure, small cities prefer to adopt local protectionist measures to prevent factor outflows; this avoids the risk that large cities will siphon off production factors, causing inefficient factor usage and increased firm-level emissions [6]. Thus, breaking down the barriers to inter-regional factor flows and increasing their rate is an effective way to reduce industrial emissions. The cross-regional flow of factors can optimize resource-allocation efficiency and upgrade industrial structures, avoiding any intensification of pollutant emissions due to mismatched resources or inappropriate industrial structures [18]. In addition, inter-regional exchanges of technology can enhance the positive effect of technology spillover. They can also encourage technology cooperation and the sharing of innovation benefits among innovation agents, thus bringing into play the energy-saving and emissions-reducing effects of technological innovation [19]. In a monocentric spatial structure, knowledge and technology diffuse slowly in regional cities, which are significantly less developed than the central city. Lin et al. (2017) argued that polycentric spatial structures can narrow regional development gaps, accelerating the flow rate of technology between neighbouring cities [20]. Similarly, the polycentric spatial structure of a region facilitates exchanges between local firms. The exchange of resources between the central city and neighbouring cities also shortens the radius of resource allocation, significantly improving production efficiency and the factor flow rate. Chen and Zhang (2021) used inter-provincial panel data as samples and intra-provincial freight volume as a proxy variable for factor flow. They showed that factor flow is an important means by which a polycentric spatial structure reduces industrial emissions [6]. Yu and Zhang (2020) conducted empirical research based on data from 28 Chinese provinces and showed that an accelerated factor flow can break regional trade barriers, optimize industrial structures, and improve the efficiency of resource use, thus reducing industrial emissions [21]. According to Chen (2020), the formation of a regional polycentric spatial structure helps to build a mutually beneficial and symbiotic development pattern among cities in the region. That study empirically verified the role of polycentric spatial structures in accelerating factor flow [22].

Second, polycentric spatial structures help increase firm size, thus reducing emissions. In a polycentric structure, industrial firms are relatively evenly distributed among cities in the region, with each city providing sufficient land, labour, and natural resources. This makes it easier to expand the scale of production [23]. The expansion of industrial firms may be accompanied by an overall increase in pollution and a lower average cost of production. When the output is constant, emissions per unit of product are reduced. In a monocentric structure, where economic activities are densely concentrated in large cities, industrial firms face fierce competition for land and labour, making it difficult to achieve economies of scale.

Finally, polycentric spatial structures can promote cooperation and the division of labour, reducing industrial emissions. Compared to monocentric structures, in which most economic activities are concentrated in a single centre, polycentric structures have lower agglomeration diseconomy effects and higher functional spillovers, deepening the labour division and cooperation between centres. Cooperation and the specialised division of labour are important ways to increase industrial efficiency and reduce emissions [24]. In a monocentric structure, production, service, and manufacturing industries are concentrated in large cities. Although smaller regional cities offer lower land prices and labour costs, manufacturing firms rarely relocate because such cities lack labour, markets, and infrastructure; they also have poor transportation [25]. Moreover, the agglomeration shadow effect caused by spatial competition drives a steady flow of people and resources from smaller cities to larger ones. A relatively balanced polycentric structure can strengthen the agglomeration-economy effect of small cities in the region. More cost-sensitive manufacturers begin to move to subcentres, while productive service industries, which are sensitive to transportation and markets, remain in large cities [26]. The labour division and spatial cooperation between the main centre and subcentres in a polycentric structure greatly enhance the productivity of individual firms, helping to reduce emissions.

Based on this analysis of the theoretical and empirical literature on the impact of polycentricity on industrial firm-level emissions, we propose the following hypothesis:

**Hypothesis** **2.**
*Urban spatial polycentricity can reduce firm-level emissions by accelerating factor flow, increasing firm size, and strengthening the division of labour.*


## 3. Model Setup and Polycentricity Index Construction

### 3.1. Model Setup

To test the hypotheses above, we drew on existing research [6] to develop the model used in this paper:(1)$$\ln\left(\mathrm{emission}\right)=\theta_{0}+\theta_{1}\ln\left(poly\right)+\underset{j}{\sum }\theta_{j}\ln\left(X_{j}\right)+\varepsilon$$
where emission represents the pollutant emissions of industrial firms, referring to the exhaust emissions of industrial firms included in this study; poly is the polycentricity index for an urban region; $X_{j}$ is a group of variables, denoting the *j*th control variable; $\theta_{1}$ and $\theta_{j}$ are the coefficient magnitudes for each corresponding variable; $\theta_{0}$ is the constant term; and ε is the residual term. A pooled cross-sectional model was used to estimate the impact of spatial structure on industry emissions.

$X_{j}$, a group of variables, includes two main categories: city-level and firm-level variables. The population size (pop), population density (popden), and GDP per capita (gdpper) are city-level variables. These are used to control the demographic and economic characteristics of a city. The city-level variables also include the main market, labour, and capital factors faced by firms. The squared term of GDP per capita (gdpper2) has been added to the model to verify the existence of an environmental Kuznets curve. The value-added share of the secondary sector of a city’s GDP (sec) reflects the industrial-development characteristics of the city as a whole. The degree of industrial development affects the intensity of local-government environmental regulations and firm-level environmental protections. Among firm-level variables, the average annual number of employees (employee) and each firm’s total industrial output value, calculated at constant prices (value), reflect the size of the firm, measured using employment and output, respectively. Other firm-level variables, i.e., the age of the firm (years), shareholdings (control), and industry category (industry) reflect growth, ownership, and industry.

Alongside the control variables, the variables ‘invest’, ‘value’, and ‘division’ are used to explore the mechanisms through which a polycentric spatial structure influences exhaust emissions. (1) Factor flow is expressed by the number of investments within an urban region (invest), i.e., the total number of times all firms invest within an urban region. The larger this variable, the more frequent the instances of capital flow and the faster the factor flow in cities within a region. (2) Enterprise size is expressed as the total industrial output value of firms, obtained from the China Industry Business Performance Database. (3) Following [27,28,29], we used the proportion of corporate managers/producers in cities, in comparison to nationwide divisional totals, to measure specialized labour categories. Given the available data, mining- and manufacturing-industry employees in each city are considered production workers, while employees in other industries are considered managers. Industry classification and employment data were obtained from the China Industry Business Performance Database.

A mixed cross-sectional data model was used for the estimation. To control for time-invariant and city-invariant factors, we added dummy variables for individual cities and years to the model. We also added the city-cluster robust standard error to mitigate the possible autocorrelation problem associated with industrial firms in the same city.

This research ran from 2000 to 2012. Raw firm-attribute data for these years were collected from the China Industry Business Performance Database, while firm-emissions data came from the China Pollution Database. The firm attributes and emission values corresponded to firm IDs, incorporating a total of 439,493 data records. City data were drawn from relevant issues of the China City Statistical Yearbook. Firm-level investment data for these years were obtained from the Qichacha.com database (accessed on 13 October 2021). The emissions of individual firms were summarized and averaged among cities. Figure 1 shows that firms in most cities had low emissions in 2000, and cities with high emissions were Panzhihua City, Jiayuguan City, Ma ‘anshan City, Tangshan City and Laiwu City. Most of these cities relied on mineral exploitation and heavy industry, resulting in high emissions. The year 2012 figures showed that the overall emissions of Chinese city firms increased, especially in the central and western regions, with the largest emissions growth in Jiayuguan City, Guigang City, Pingliang City, Karamay City and Handan City. As can be seen from Figure 1, from 2000 to 2012, the emissions of industrial firms in Chinese cities increased significantly, especially in the central and western regions.

### 3.2. Polycentric Index Construction

Polycentric development balances the importance of cities within a given region [30]. It does not simply consider ‘cities’ that coexist or interact in a region. In general, polycentricity has both morphological and functional dimensions [31]. Morphological polycentricity, the focus of this study, deals with the balanced distribution of urban importance within a region. In most studies, this importance is measured using employment, population size, and economic activities. Three empirical approaches are used to measure polycentricity:The slope of the rank-size distribution [32] based on the statistical regression line between the size and corresponding ranks of each subunit within an urban region;The share of the centre of an urban region occupied by employment subcentres [33] or the number of extra subcentres [34];Primacy [35], or the extent to which the largest employment centres deviate in size from the regression line through a rank-size distribution.

In contrast to the latter two approaches, the rank-size method has been used in many studies [36,37]. Importantly, the widely accepted rank-size-based Pareto exponential distribution can be used to determine whether employment centres are concentrated within or scattered outside the main centre, the key aim of the present study.

Here, we measured the polycentricity of Chinese urban regions at or above prefecture-level, via the rank-size-based Pareto exponential distribution. In the Chinese administrative system, an urban region is an administrative area consisting of a city proper (metropolis), counties, and county-level cities, each of which is referred to as a ‘subunit’. According to [38], the rank-size order of the largest centres may indicate not simply an urban system with one dominant centre (a somewhat monocentric situation), but also a polycentric system of similarly sized cities that lack a strong hierarchy. *Rank*-1/2, rather than the actual ‘rank’, has been used to reduce the bias associated with small sample sizes [39]. Urban regions with one or no subunits were removed from the dataset [40]. The specific calculation method involved regressing the size of all subunits in an urban region (as the explanatory variable) against their rankings, as the coefficient of size was the Pareto exponential distribution. The polycentricity index was computed as follows:ln(Rank − 1/2) = *a* + bln(*Size*) *+*
*ε*(2)
where *b* is the Pareto exponential distribution or the level of morphological polycentricity (*poly*), such that a high *b*-value implies a high level of polycentricity. In addition, *Rank* and *Size* refer to the subunit employment rank and employment size (i.e., the city proper, counties, and county-level cities in the urban region) at prefecture level and above urban regions; *a* and ε are the constant and error terms, respectively. Employment data were obtained from the China Industry Business Performance Database. Table 1 shows the statistical characteristics of the variables used in this study.

## 4. Empirical Tests and Analysis

### 4.1. Baseline Regression and Causal Identification

City-cluster robust standard errors were applied to mitigate the autocorrelation of industrial firms within the same city (e.g., mutual learning among firms in the same city has an impact on emissions). Table 2 reports the results of the baseline regressions. Columns (1), (2) and (3) show the results of univariate regressions (controlling for the polycentricity index only), regressions controlling for city-level variables, and regressions of both city- and firm-level variables. The results of the three regressions show that the polycentricity index (poly) is significantly negative. This indicates that the more polycentric a city structure is, the lower the exhaust emissions from industrial firms. In other words, polycentricity helps firms reduce emissions. This result does not change as the variable increases or decreases.

The remaining explanatory variables are described below. The higher a city’s population density, the lower the firm-level emissions; this may be due to agglomeration economies. The negative coefficient of the squared term of GDP per capita indicates that, as GDP per capita increases, firm-level emissions tend to rise and then fall, in line with the environmental Kuznets curve effect. For a firm, more employees and higher gross output indicate higher emissions, which seems to contradict the accepted expectation of economies of scale. This is possibly because there is a threshold at which firms achieve economies of scale and most firms in Chinese cities are too small to reach that threshold. We attempted to add a quadratic term for scale to the model, but it was not significant. Thus, the quadratic term for scale was not controlled in the model. The longer a firm is established, the lower its emissions become. This is probably because the longer a firm is established, the more it accumulates capital and technology and invests in environmental protection. In addition, older firms are subject to longer and more intense environmental regulation by local governments.

The spatial structure of cities influences firm emissions. Conversely, the emissions behaviour of firms affects the spatial layout of cities. Sufficient evidence has shown that local governments are incentivized to decentralize pollution-intensive companies, such as by levying taxes in their own city and transferring carbon emissions and pollutants to neighbouring cities [41,42,43]. We drew on the approach of [9] to address the estimation bias and inconsistency caused by reverse causality. In particular, the Gini coefficient (built_gini) of the built-up areas of municipalities, counties, and county-level cities in an urban region was used as an instrumental variable for urban spatial structure, while a two-stage least squares (TSLS) regression was used to test the endogeneity of the model. This instrumental variable is appropriate for two reasons. First, the distribution of built-up areas within an urban region directly influences the spatial distribution of industrial firms. By influencing the size of the built-up area, a land policy that coordinates and balances inter-regional construction land targets limits the economic growth space and the number of people the city can accommodate, thus influencing the spatial concentration of population and firms. Second, the distribution of built-up areas within an urban region is exogenous to the pollutant emissions of firms. On the one hand, the spatial distribution of construction land can affect the urban environment only through the behaviours of the subjects (such as industrial firms) based on it. The amount of construction land does not directly affect the production emissions of industrial firms. On the other hand, the current land planning outlines of Chinese cities do not take environmental pollution, especially industrial emissions, as influencing factors to be directly considered in the allocation of construction land indicators. In other words, urban industrial emissions are not within the direct target range of construction land planning. Lu and Feng (2014), after streamlining the institutional context of China’s current land policy, argued that environmental pollution, especially industrial pollutant emissions, was not an influencing factor in the allocation of construction land targets [9]. In other words, the increase, decrease, and distribution of built-up areas (construction land) in cities were not directly related to the emissions of industrial firms.

As for the potential for weak instrumental bias, the Cragg-Donald Wald F statistic of these models is 22.075, which is greater than the empirical value of 10, suggesting that there is no serious weak instrumental variable bias. Second, the F statistic is 22.08 (greater than 10), and its *p* value is 0.000, indicating that there is little need to worry about weak instrumental bias. Third, according to a 5% Wald test of ‘nominal size’ of endogenous explanatory variables, at the ‘true size’ of 15%, the minimum eigenvalue statistic (F statistic) is greater than the corresponding critical value (4.58); the null hypothesis of ‘weak instrumental variable’ can, therefore, be rejected. Fourth, we used the limited information maximum likelihood (LIML) estimation, which is not sensitive to weak instrumental variables, to test the model. By comparison, the coefficients and significance of LIML are very close to those of 2SLS, which also proves that ‘there is no weak instrumental bias’. Table 3 lists the results of the two-stage estimation of instrumental variables. It can be seen from the first-stage results that built_gini is significantly negatively correlated with the polycentricity index (poly). This indicates that the more concentrated the distribution of built-up areas in a city (the larger the coefficient of built_gini), the more monocentric the structural distribution of industrial firms will be within that city (the smaller the poly coefficient). According to the second-stage results, the polycentricity index is significantly negative, which is consistent with the aforementioned findings of this study.

### 4.2. Robustness Tests

If researchers assess the level of polycentricity in an urban region without a clear rationale for how many centres to include, their measurement results may be shaped by the pre hoc selection of centres [44]. In the administrative divisions of China, different urban regions contain different numbers of spatial subunits. To achieve comparability among Pareto exponential distributions in these urban regions, Meijers (2008) [45] and Sun et al. (2020) [46] argued that polycentricity calculations should be based on the same number of spatial subunits in different urban regions. We computed the Pareto exponential distribution of the top three, six, and ten (top3, top6 and top10) spatial subunits in each urban region, in accordance with the tertiles of spatial subunits in all urban regions. This allowed us to analyse the robustness of the core independent variable. The specific method employed involved using the top three subunits in each urban region, ranked by employment size, to calculate the Pareto exponential distribution via Equation (1); this is referred to as ‘polycentricity_top3’ in the subsequent tables. Where there were fewer than three subunits in an urban region, all subunits were used instead. The ‘polycentricity_top6’ and ‘polycentricity_top10’ were calculated in the same way as the ‘polycentricity_top3’. In addition, the Primacy index describes the ‘relative importance’ of the first city in a region. The Hirschman–Herfindahl index (HHI) is the sum of the squared shares of each subunit in a region, often used to measure the concentration of industries. These two indices are generally used to characterize the spatial structure of a region. The measures of these two indices can be found in Sun et al. (2019) [47]. The larger the two indices, the weaker the polycentricity in an urban region. In Table 4, all of the polycentricity indices are shown to be negatively correlated with industrial firm-level emissions; this is consistent with the results shown in Table 1 and Table 2.

Different industries have different potential pollutant emissions. As they are closely connected with production, manufacturing firms emit more pollutants. Productive-service industries are closely connected with the consumption side and emit fewer pollutants. The lack of industry-specific tests may result in biased conclusions. For this reason, we classified all samples into three industry categories: mining industries; manufacturing industries; and electricity, heat, gas, and water production and supply industries, in accordance with the Industrial Classification for National Economic Activities (GB/T 4754-2017). A regression analysis was conducted for each of the three industry categories; the results are shown in Table 5. Clearly, among mining and manufacturing firms, an urban polycentric structure can help to reduce the emissions of individual firms. For firms in the electricity, heat, gas, and water production and supply industries, there is no evidence that polycentricity affects emissions. When it comes to industrial structure, the emission-reduction effect of polycentric spatial structure on firms derives mainly from the mining and manufacturing industries.

Urban spatial structure depends on city size. For this reason, different spatial structures may be applied to cities of different sizes to achieve emission reductions. Zhang and Derudder (2019) found that measures of spatial structure, in particular monocentric and polycentric measures, were sensitive to geospatial scale [44]. Han et al. (2020) observed that monocentric structures could help small cities reduce PM _2.5_ concentrations, while polycentricity could help large cities reduce PM_2.5_ concentrations [17]. In a study of the Yangtze River Delta, Li and Phelps (2018) argued that the degree of polycentricity could change as the study area expanded and the number of cities increased [48]. This was because an increase in the study area could bring more small- and medium-sized cities into the sample, affecting the balance between cities. For this reason, we divided the sample cities by population size into three groups: small cities (population ≤ 3,729,600), medium-sized cities (3,729,600 < population ≤ 6,478,000), and large cities (population > 6,478,000). Table 6 presents the subsample regression results. It can be seen that, in medium-sized and large cities, the polycentric structure significantly promotes the emission reductions of firms, while in small cities, no emission-reduction effect of polycentricity is found. This indicates that the emission-reduction effect of polycentric spatial structure mainly comes from medium-sized and large cities.

## 5. Mechanisms

To examine how polycentricity can reduce emissions from industrial firms, we designed three pathways based on previous studies and our conceptual hypotheses. First, polycentric cities can improve firm resource use by accelerating factor flow; in this way, they can reduce emissions. The interaction term between the polycentricity index and the average investment amount was added to the model. As shown in column (17) of Table 7, the coefficient of this interaction term is significantly positive, indicating that the emission-reduction effect of intra-city investment amounts on industrial firms increases with the polycentricity index. In other words, the polycentric spatial structure can strengthen the emission-reduction effect of urban investment on industrial firms. Second, the polycentric structure helps to increase firm size, reducing emissions through economies of scale. The interaction term between the polycentricity index and the total firm output value was added to the model, and the results are shown in column (18) of Table 7. As the interaction term is significantly positive, the emission-reduction effect of firm size on industrial firms increases with the polycentricity index. In other words, the polycentric spatial structure can increase the emission-reduction effect of firm size on industrial firms. The effect of gross firm output on firm-level emissions is positive in column (3) of Table 2 and negative in column (18) of Table 7. This difference precisely illustrates the need for an interaction term, suggesting that the effect of firm size on emissions is mediated by spatial structure. Finally, polycentricity contributes to the division of labour in regional cities, while specialized labour division facilitates more efficient production and lower emissions. The dependent variable of the baseline model was replaced by the labour division index (*division*). As shown in column (19) of Table 7, the coefficient of the polycentricity index is significantly positive; as urban regions develop a polycentric structure, the division of labour in the city grows more specialized. This study confirms that a specialized division of labour is significantly negatively correlated with firm-level pollutant emissions.

## 6. Conclusions

The environmental sustainability of urban structures continues to attract increasing interest, especially from government agencies aiming to reduce industrial firm-level emissions at source via polycentric spatial structures. Based on a pooled cross-sectional model of the exhaust emissions of individual industrial firms, this study contributes to the current debate by empirically analysing the emissions data and impact mechanisms of individual firms. By applying an instrumental variable model and robustness tests, we identified a significantly negative correlation between urban-region polycentricity and firm-level emissions. This finding supports the results of previous studies, providing a quantitative explanation of accelerated factors, larger firm size, and enhanced specialization in the division of labour. Quantitative analyses revealed that a polycentric structure reduces industrial firm-level emissions in urban regions. After a series of robustness tests, the research conclusions remained unchanged.

Our findings have important policy implications. First, the results indicate that spatial structure has an impact on firm-level emission reductions; thus, the optimization of urban spatial structure is a promising policy tool, able to improve environment quality. Second, a polycentric spatial structure can enhance the emission-reduction performance of factor flows, firm size, and specialized labour division. City planners should consider industrial policies when planning polycentric spatial structures for medium-sized and large cities. In addition, they should strengthen population and capital flows, encourage the development of large firms, and cultivate industries with urban characteristics to achieve a specialized division of labour, which will help firms reduce emissions.

However, these results should be interpreted with some caution. On the one hand, one contribution of this study is the use of firm-level emissions data. Due to data limitations, the firm-year panel data could not be used to track interannual variations in the emissions of individual firms. Although we controlled for city and year fixed effects in the pooled cross-sectional model, firm-level fixed effects could not be eliminated. As a result, some time-invariant firm factors may not have been effectively controlled for, leading to estimation bias. On the other hand, the change in urban spatial structure is relatively slow, and long-term series data can capture the change in spatial structure in time. Future work should focus on sourcing qualified employment spatial data and firm-level emissions data for an accurate estimation of polycentric environmental performance.

## Figures and Tables

**Figure 1 ijerph-19-08167-f001:**
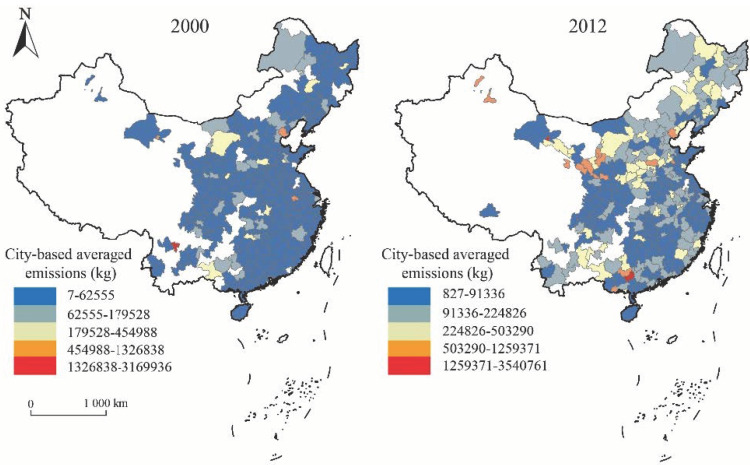
Spatial distribution of emissions from Chinese industrial firms in 2000 (**left**) and 2012 (**right**).

**Table 1 ijerph-19-08167-t001:** Descriptive statistics of variables.

Variable	Description	Obs	Mean	Std. Dev.	Min	Max
ln(emission)	Exhaust emissions of industrial firms	297,259	8.005	2.338	−4.605	18.52
ln(poly)	Polycentricity index	307,846	−0.678	0.624	−2.035	7.064
ln(pop)	Population size	435,569	6.187	0.663	2.770	8.115
ln(popden)	Population density	435,569	−3.074	0.783	−7.663	0.145
ln(gdpper)	The GDP per capita	434,861	10.23	0.930	7.401	13.02
ln(sec)	The value-added share of the secondary sector of a city’s GDP	435,569	3.902	0.193	0.978	4.511
ln(employee)	Average annual number of employees	431,519	5.506	1.178	0	13.23
ln(value)	Firm’s total industrial output value	435,134	10.87	1.836	−2.303	19.21
ln(year)	Age of the firm	434,312	2.199	0.895	0	7.606

**Table 2 ijerph-19-08167-t002:** Baseline results.

Dependent Variable: ln(Emission)	(1)	(2)	(3)
ln(poly)	−0.035 *	−0.035 *	−0.031 **
	(0.021)	(0.021)	(0.014)
ln(pop)		0.287	0.030
		(0.211)	(0.141)
ln(popden)		0.027	−0.096 *
		(0.055)	(0.047)
ln(gdpper)		0.068	0.652
		(0.722)	(0.487)
ln(gdpper2)		0.007	−0.044 **
		(0.029)	(0.020)
ln(sec)		−0.238	−0.228
		(0.273)	(0.226)
ln(employee)			0.382 ***
			(0.019)
ln(value)			0.439 ***
			(0.011)
ln(year)			−0.045 ***
			(0.010)
Dummy (control)			Yes
Dummy (industry)			Yes
City fixed effect	Yes	Yes	Yes
Year fixed effect	Yes	Yes	Yes
_cons	7.897 ***	5.98	−2.760
	(0.077)	(4.132)	(5.715)
Observations	241,475	241,151	233,383
R-squared	0.0834	0.0836	0.4822

Note: ***, **, and * are the significance at the 1%, 5% and 10% levels, respectively. City-cluster robust standard errors are in parentheses.

**Table 3 ijerph-19-08167-t003:** IV results.

First-Stage Results		Second-Stage Results	
	(4)		(5)
Dependent variable	ln(poly)	Dependent variable	ln(emission)
ln(built_gini)	−0.013 ***	ln(poly)	−0.508 **
	(0.003)		(0.240)
Other variables	Yes	Other variables	Yes
City fixed effect	Yes	City fixed effect	Yes
Year fixed effect	Yes	Year fixed effect	Yes
Observations	233,383	Observations	233,383
F-statistic	22.08	Cragg-Donald Wald F statistic	22.075
Prob > F	0.000	R-squared	0.475

Note: ***, **, and * are the significance at the 1%, 5% and 10% levels, respectively. Robust standard errors are in parentheses.

**Table 4 ijerph-19-08167-t004:** Tests for polycentricity robustness.

Dependent Variable: ln(Emission)	(6)	(7)	(8)	(9)	(10)
ln(poly_top3)	−0.025 *				
	(0.014)				
ln(poly_top6)		−0.033 **			
		(0.014)			
ln(poly_top10)			−0.031 **		
			(0.014)		
ln(primacy)				0.035 **	
				(0.019)	
ln(HHI)					0.018 **
					(0.0106)
Other variables	Yes	Yes	Yes	Yes	Yes
Observations	233,269	232,580	232,290	233,700	233,700
R-squared	0.482	0.482	0.482	0.483	0.482

Note: ***, **, and * indicate significance at the 1%, 5% and 10% levels, respectively. City-cluster robust standard errors are in parentheses.

**Table 5 ijerph-19-08167-t005:** Tests for different industries.

Dependent Variable: ln(Emission)	(11)	(12)	(13)
	Mining Industries	Manufacturing Industries	Electricity, Heat, Gas, and Water Production and Supply Industries
ln(poly)	−0.106 *	−0.029 **	−0.027
	(0.060)	(0.015)	(0.034)
Other variables	Yes	Yes	Yes
Observations	9013	218,635	5732
R-squared	0.443	0.453	0.672

Note: ***, **, and * indicate significance at the 1%, 5% and 10% levels, respectively. City-cluster robust standard errors are in parentheses.

**Table 6 ijerph-19-08167-t006:** City-size tests.

Dependent Variable: ln(Emission)	(14)	(15)	(16)
	Small Cities	Medium-Sized Cities	Large Cities
ln(poly)	−0.030	−0.046 **	−0.005 **
	(0.023)	(0.022)	(0.002)
Other variables	Yes	Yes	Yes
Observations	64,298	82,586	86,566
R-squared	0.527	0.472	0.464

Note: ***, **, and * indicate significance at the 1%, 5% and 10% levels, respectively. City-cluster robust standard errors are in parentheses.

**Table 7 ijerph-19-08167-t007:** Results of the mechanism analysis.

	(17)		(18)		(19)	
Dependent Variable	ln(Emission)		ln(Emission)		ln(Division)	ln(Emission)
ln(poly)	−0.043 **	ln(poly)	−0.107 *	ln(poly)	0.133 *	
	(0.019)		(0.009)		(0.100)	
ln(invest)	−0.007 *	ln(value)	−0.443 ***	ln(division)		−0.169 **
	(0.004)		(0.000)			(0.017)
ln(poly) × ln(invest)	0.002 **	ln(poly) × ln(value)	0.007 **			
	(0.001)		(0.003)			
Other variables	Yes	Other variables	Yes	Other variables	Yes	Yes
Observations	221,753	Observations	233,383	Observations	20,141	20,141
R-squared	0.475	R-squared	0.482	R-squared	0.931	0.374

Note: ***, **, and * indicate significance at the 1%, 5% and 10% levels, respectively. City-cluster robust standard errors are in parentheses.

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
