# Peer review of "Does a Polycentric Spatial Structure Help to Reduce Industry Emissions?"

_ijerph, 2022, doi:10.3390/ijerph19138167_

Round 1

Reviewer 1 Report

Dear Authors,

I have the following comments to your manuscript.

Page 1, lines 5, 6, 7

The address of your institutions, please write clearly with the full addresses of institutions involved and always separated by new line

page 3, line 98, 99

the sentence " In summary, industrial emissions are likely to increase as agglomeration deepens."

the meaning is not clear, please rewrite it.

Page 8, line 314

the sentence "Conversely, the emissions behaviour of firms affects the spatial layout of cities. sufficient evidence has shown that local governments are incentivized to decentralize pollution-intensive companies, such as by levying taxes in their own city and transferring carbon emissions and pollutants to neighboring cities "

Do you think that " transferring carbon emissions and pollutants to neighboring cities " is the way haw to reduce the industry emissions ?

The emissions are toxic gases, particulates and other form of toxic materials.

Reduction of emission from scientific point is reduction by proper way e.g. by filtering of toxic emissions, collecting toxic emissions ... etc. and not by transferring of emission to other city....

otherwise you could title your manuscript instead of "Does a polycentric spatial structure help to reduce industry emissions?

" Does a polycentric spatial structure help to transfer industry emissions? From city A to city B." Would you think that this is appropriate?

Page 12, line 469

the sentence " The conclusions remain robust after various tests, including changing the polycentricity index measure, the type of industry in the sample, and city size. "

Try to rewrite this sentence with better explanation of your thoughts. Explain it with more details.

I would suggest you to improve the chapter "4. Empirical tests and analysis" and chapter "5. Mechanisms", the way to be more interesting for readers.

Try to use pictures and graphs if possible.

Regarding chapter "6. Conclusions"

Here I would suggest you to clearly summarise your main research ideas, main results and include some motivation for the future fork. 

Author Response

Reviewer 1

Dear Authors,

 I have the following comments to your manuscript.

  • Page 1, lines 5, 6, 7

The address of your institutions, please write clearly with the full addresses of institutions involved and always separated by new line.

 Thanks to the reviewer's suggestion, we have made the corresponding changes.

  • page 3, line 98, 99

the sentence " In summary, industrial emissions are likely to increase as agglomeration deepens." the meaning is not clear, please rewrite it.

 Thanks to the reviewer's suggestion, we have made the corresponding changes.

  • Page 8, line 314

the sentence "Conversely, the emissions behaviour of firms affects the spatial layout of cities. sufficient evidence has shown that local governments are incentivized to decentralize pollution-intensive companies, such as by levying taxes in their own city and transferring carbon emissions and pollutants to neighboring cities " Do you think that " transferring carbon emissions and pollutants to neighboring cities " is the way haw to reduce the industry emissions ?

 The emissions are toxic gases, particulates and other form of toxic materials.

Reduction of emission from scientific point is reduction by proper way e.g. by filtering of toxic emissions, collecting toxic emissions ... etc. and not by transferring of emission to other city....

 otherwise you could title your manuscript instead of "Does a polycentric spatial structure help to reduce industry emissions?

" Does a polycentric spatial structure help to transfer industry emissions? From city A to city B." Would you think that this is appropriate?

Thanks to the reviewers for their suggestions. The purpose of this paragraph is to discuss the bidirectional relationship between urban spatial structure and enterprise emissions, and the resulting reverse causal bias. On the one hand, urban spatial structure will affect enterprise emissions, on the other hand, the emission behavior of industrial firms will also affect urban spatial structure. For example, pollution-intensive industrial firms tend to be located at the edge of the city, which leads to the polycentricity. If we do not remove the effect of corporate emissions on spatial structure in the regression analysis, the results in the manuscript may be biased and unreliable.

Therefore, the purpose of this paragraph is not to explain "transferring carbon emissions and pollutants to neighboring cities is the way haw to reduce the industry emissions", but to argue the possible bidirectional relationship between spatial structure and corporate emissions. We use instrumental variables method to eliminate the possible reverse causality, and has identified the one-way causal relationship between spatial structure and enterprise emissions.

  • Page 12, line 469

the sentence " The conclusions remain robust after various tests, including changing the polycentricity index measure, the type of industry in the sample, and city size. " Try to rewrite this sentence with better explanation of your thoughts. Explain it with more details.

Thank you for your detailed suggestions. Modify the original sentence to "After a series of robustness tests, the research conclusions remain unchanged". 

  • I would suggest you to improve the chapter "4. Empirical tests and analysis" and chapter "5. Mechanisms", the way to be more interesting for readers. Try to use pictures and graphs if possible.

Thank you for your suggestions. From the perspective of the research paradigm of economics, regression results all are presented in the form of tables. The advantage lies in the intuitive visibility of variable results and significance. We are also exploring the use of diagrams to show results.

  • Regarding chapter "6. Conclusions". Here I would suggest you to clearly summarise your main research ideas, main results and include some motivation for the future fork. 

The reviewer's suggestion is excellent. We have re-examined the Conclusions section and revised accordingly as requested by the reviewers.

Reviewer 2 Report

Title should be changed to Reducing Environmental Pollution by Polycentric Space Development
given that from the current title it is inferred that the production of pollutants by the industry is reduced, something that will not be verified until analyzing the data of the industry year

Author Response

Reviewer 2

Title should be changed to Reducing Environmental Pollution by Polycentric Space Development given that from the current title it is inferred that the production of pollutants by the industry is reduced, something that will not be verified until analyzing the data of the industry year

The reviewer's suggestion is great! The research question of this paper is: to explore whether and how urban polycentric spatial structure affects corporate emissions? Existing studies have not reached consistent conclusions. We provide emission data based on individual enterprises to test the emission reduction performance of the polycentric structure.

Therefore, replacing the title with “Reducing Environmental Pollution by Polycentric Space Development” may help reveal the research results, but it lacks Space for readers to think and makes the reading less interesting.

Reviewer 3 Report

The topic is interesting and the paper is written in a very easy way to be followed and observed. More recent relevant papers in literature review could be useful. Also, some details on the relevance of the results, considering that the analysis is based mainly on the Chinese data and the topic might prove interesting at global level. The means to overcome the limits of research explained in conclusion could be more detailed.

Author Response

Reviewer 3

The topic is interesting and the paper is written in a very easy way to be followed and observed. More recent relevant papers in literature review could be useful. Also, some details on the relevance of the results, considering that the analysis is based mainly on the Chinese data and the topic might prove interesting at global level. The means to overcome the limits of research explained in conclusion could be more detailed.

Thanks to reviewers for their praise and suggestions. We re-examined the manuscript and revised the conclusion.
